# Explaining recommendation systems through contrapositive perturbations

## Abstract

Recommender systems are widely used to help users discover new items online. A popular method for recommendations is factorization models, which predict a user's preference for an item based on latent factors derived from their interaction history. However, explaining why a particular item was recommended to a user is challenging, and current approaches such as counterfactual explanations can be computationally expensive. In this paper, we propose a new approach called contrapositive explanations ($\mathcal{Contra+}$ ) that leverages a different logical structure to counterfactual explanations. We show how contrapositive explanations can be used to explain recommendation systems, by presenting a methodology that focuses on finding an explanation in the form of "*Because the user interacted with item j, we recommend item i to the user*," which we show is easier to compute and find compared to traditional counterfactual approaches which aim at "*Because the user **did not** interacted with item j, we **did not** recommend item i to the user*,". We evaluate our approach on several real-world datasets and show that it provides effective and efficient explanations compared to other existing methods.

## 1 Introduction

Recommender systems have become ubiquitous in online platforms to help users discover new items of interest Lü et al. (2012); Aggarwal et al. (2016); Beel et al. (2016); Jannach et al. (2022). These systems analyze a user's historical interactions with items and suggest new items based on those interactions to provide personalized recommenders that align with the user's preferences Lu et al. (2015); Das et al. (2017); Bobadilla et al. (2013); Pazzani & Billsus (2007). Factorization models, such as for example the Singular Value Decomposition (SVD) model, are commonly used in recommender systems Guan et al. (2017); Bokde et al. (2015) to predict a user's preference for an item based on latent factors derived from the user's interaction history.

However, the "*why*" behind a recommendation remains a challenging issue. Counterfactual explanations Wachter et al. (2017) offer one possible approach to this problem; they attempt to demonstrate the minimal changes needed in a user's history that would trigger a different recommendation Tran et al. (2021). This, however, requires the deletion of the (user, item) pair from the user's history and retraining of the model, a process that is time-consuming and computationally expensive.

In order to bridge this gap, various techniques have been proposed, such as influence functions Tran et al. (2021); Koh & Liang (2017). Despite their utility in computing the impact of data removal, these methods are still challenged by their computational demands, primarily due to the need to compute the inverse of the Hessian. This high computational cost often limits their practical application in real-time recommender systems; which is the primary focus of this paper. In addition to that, influence functions are only approximations and hence are less reliable when it comes to highly non-linear models such as Deep Neural Networks Basu et al. (2020).

To address this issue, this paper introduces a novel approach called *contrapositive explanations* ($\mathcal{Contra+}$ ). The contrapositive logic involves negating and switching the order of the antecedent and consequent of an implication statement. The proposed approach in this paper leverages this logic to provide explanations for recommenders by first negating the user's recommended item and switching it with another item and secondly inspecting the resulting changes in the user's history. This approach avoids the need for retraining the model and provides a more efficient way to generate

explanations. Before diving deeper into how we can utilize this logic in recommender systems, let us take a short detour to lay out what the contrapositive logic entails in a simple example.

**Example 1.1.** *[Toy Example] Consider the following two statements:*

- *A: It is raining.*

- *B: The road is wet.*

From the above, we can make the following logical statements: $A \rightarrow B$ i.e. It was raining and this implies that the road is wet. The logical equivalent is $\bar{B} \rightarrow \bar{A}$ i.e. The road is **not** wet which implies that it was **not** raining. This is contrary to the counterfactual logic which would reason through $\bar{A} \rightarrow \bar{B}$ i.e. It was **not** raining and this implied that the road is **not** wet. Note, that this is not always the case, as a bucket of water could make the road wet. Hence, these are two distinct statements and in this paper, in particular, we focus on trying to achieve $A \rightarrow B$.

With this in mind, we now present how we can use contrapositive logic in explaining recommender systems. The explanation logic that we will be using throughout the paper is the following:

**Example 1.2.** *[Recommender System Example] Consider again the following two statements:*

- *A: The user $u$ interacted with item $j$ in the user history.*

- *B: The user $u$ is recommended item $i$.*

Here, the objective is to find an explanation that supports the statement $A \rightarrow B$, meaning because user $u$ interacted with item $j$, item $i$ was recommended. Identifying such explanations can be challenging and computationally intensive, as it would require exhaustively searching through all possible combinations of a user's history to determine which interactions, when removed, do not alter the recommendation. To address this challenge, we adopt the logically equivalent contrapositive route $\bar{B} \rightarrow \bar{A}$: if item $i$ is not recommended, then user $u$ would not have interacted with item $j$.

Intuitively, given the predominance of user and item embeddings in most recommender systems, our method starts by invoking $\bar{B}$, that is, we "*perturb*" the user embedding to ensure item $i$ is not recommended. Then, given this perturbed user embedding, we identify the historical item that has lost most relevance to the user — effectively, the item with which the user would not have interacted, denoted as $\bar{A}$. We detail the formalization of our method in Section 3.

The key contributions of this paper are as follows:

- We propose an explanation method for recommender systems that uses contrapositive logic, which involves negating and switching the antecedent and consequent of a user's preference for items. This approach reduces the computational cost and the need for model retraining.
- We propose a computationally efficient framework recommender system. Specifically, we investigate its applicability and performance on SVD and MLP-based recommender systems, demonstrating its versatility in various experiments.
- We introduce an evaluation metric tailored for contrapositive logic, offering a new perspective to assessing explanations compared to traditional counterfactual logic. We demonstrate on extensive experiments that our proposed method is able to outperform existing methods.

This paper is structured as follows: Section 2 gives background on recommender systems and existing explanation methods. Section 3, introduces our proposed methodology $\mathcal{C}ontra+$ , which is then followed by extensive experiments in section 4. Lastly, in section 5, we conclude with the limitations as well as future extensions to our proposed method $\mathcal{C}ontra+$ .

## 2 Background and Related Work

### 2.1 Formulation of Recommender Systems

Before diving into the specifics of Singular Value Decomposition (SVD) and Multi-Layer Perceptron (MLP) models, we first establish the fundamental elements of recommender systems. The key components for SVD and MLP models are as follows:

- **User-Item pair** $(u, i)$**:** These pairs represent the interaction between user $u \in \mathcal{U}$ and item $i \in \mathcal{I}_u$, where $\mathcal{U}$ is the set of all users and $\mathcal{I}_u$ is the set of items user $u$ has interacted.

- **Training data:** The data in recommender systems usually comprises of a user-item interaction matrix $R \in \mathbb{R}^{m \times n}$, with $m$ representing the number of users and $n$ the total number of items. Each element $R_{ui}$ corresponds to the rating given by user $u$ to item $i$.

- **User/Item embeddings** $(p_u, q_i)$**:** Each user $u$ and item $i$ are represented in a latent space through vectors, or embeddings, denoted as $p_u$ and $q_i$ respectively. These embeddings are computed during the training process (SVD or MLP) and capture the underlying characteristics and preferences of users and items.

## 2.2 BRIEF OVERVIEW OF SVD AND MLP MODELS

**Singular Value Decomposition:** SVD is a widely used matrix factorization Bokde et al. (2015) model in recommender systems and allows us to predict user preferences by decomposing the user-item interaction matrix $R$ into two low-rank matrices: $P \in \mathbb{R}^{m \times d}$ and $Q \in \mathbb{R}^{n \times d}$, according to:

$$R \approx PQ^T \tag{1}$$

Here, $d$ is the pre-determined number of latent factors. Each row in the matrices $P$ and $Q$ represents a latent factor vector for a user and an item, respectively. These vectors, denoted as $p_u$ and $q_i$, serve as embeddings that encapsulate the essential characteristics of user $u$ and item $i$ in a $d$-dimensional space. To leverage the predictive power of SVD models, we first compute an interaction score between a user and every non-rated item. This interaction score signifies the predicted rating or preference of user $u$ for item $i$ and is calculated as the dot product of the corresponding user and item embeddings. Consequently, the score function $s(u, i)$ is defined as:

$$s(u, i) = p_u^T q_i = \langle p_u, q_i \rangle \tag{2}$$

Once, we have computed the scores between the user $u$ and all non-rated items, we can then sort the score and recommend the item which gave us the highest interaction score.

**Multi-Layer Perceptron:** On the other hand, MLP models extend beyond linear relationships captured by SVD. They leverage neural networks to process the concatenated user and item embeddings, thereby capturing potential non-linear interactions between users and items. In this case, given the user embedding $p_u$ and the item embedding $q_i$, the score function, denoted as $s(u, i)$, is defined as:

$$s(u, i) = \texttt{MLP}([p_u; q_i]; \theta) \tag{3}$$

Here $\texttt{MLP}([p_u; q_i]; \theta)$ is a neural network parameterized by $\theta$. Similarly to the SVD model, when making a new recommendation for the user, we sort the score and pick the highest-scored item. Next, we delve deeper into the specifics of these models and how explanations can be generated.

## 2.3 COUNTERFACTUAL EXPLANATIONS AND INFLUENCE FUNCTIONS

One of the primary ways of explaining recommender systems is through counterfactual explanations Wachter et al. (2017); Tran et al. (2021); Yao et al. (2022a); Ghazimatin et al. (2020); Kaffes et al. (2021); Tan et al. (2021). These methods attempt to compute logical statements of the form $\bar{A} \rightarrow \bar{B}$, which indicates that because the user did not interact with item $j$, item $i$ was not recommended. In other words, because the removal of item $j$ changed the recommendation for user $u$, item $j$ serves as an explanation for having had an impact on the recommendation of item $i$.

However, computing such counterfactual explanations can be challenging, particularly when attempting to identify which historical item(s) are responsible for a given recommendation. One approach is to remove a combination of relevant item(s) from the training data and retrain the model, but this can be computationally infeasible, particularly for large neural networks. To address this issue, researchers have proposed alternative methods such as gradient-based Tan et al. (2021) and search-based Kaffes et al. (2021) approaches, as well as influence functions Tran et al. (2021); Koh & Liang (2017), that approximate the retraining of a model when one or more data points are removed. However, even though these methods significantly reduce the computational cost of retraining models, these methods are not suitable for large real-time recommender systems given that they still require significant computational requirements.

In particular, influence functions have become popular due to their ability to approximate a re-trained differentiable model without requiring the retraining of the entire model. However, they are not without their own limitations, such as the need to compute the Hessian matrix, which can be computationally infeasible for large networks, and their second-order approximation of the model, which can result in misleading results Basu et al. (2020). Others have tried to train surrogate models that learn a mapping from removed items to retrained models Yao et al. (2022a). However, the latter suffers from extensive offline training of the surrogate model which can be prohibitive in practice.

To address these challenges, we propose a new approach to explainable recommendation systems based on contrapositive explanations. Unlike counterfactual explanations, which attempt to identify the necessary cause of a recommendation, contrapositive explanations focus on identifying the sufficient conditions for a recommendation to be made. Specifically, we aim to compute logical statements of the form $\bar{B} \to \bar{A}$, which is equivalent to $A \to B$.

# 3 PROPOSED METHOD: $\mathcal{C}ontra+$ EXPLANATIONS

In this section, we introduce our novel approach for generating what we term $\mathcal{C}ontra+$ explanations for any recommender system. We start by focusing on the SVD model as a simple case study and later explain how to apply our proposed method to any differentiable model such as MLPs.

## 3.1 FACTOR MODEL: SVD

Recall that the SVD model is a factorization-based approach that represents users and items in a shared latent space. A rating for user-item pair $(u, i)$ is predicted using a factor model, where the interaction between the user and item is represented $s(u, i) = \langle p_u, q_i \rangle$, where $p_u, q_i \in \mathbb{R}^d$ are $d$-dimensional latent factors that measure the alignment between the preferences of user $u$ and the item $i$. Our goal is to arrive at the statement $\bar{B} \to \bar{A}$, which means that: Because we do **not** recommend item $i$ to user $u$, the user would **not** have interacted with item $j$.

As a first step, we start by negating the consequent: $B$ : "*we recommend item $i$ to user $u$*". Our method $\mathcal{C}ontra+$ first constructs a user embedding $p'_u$ for user $u$ such that item $i$ is not recommended. To achieve this, we perturb the user's latent representation $p_u$ in the opposite direction to the item's representation $q_i$, such that the recommendation score decreases and item $i$ is no longer recommended. In other words, we enforce $\bar{B}$, the negative for "*user $u$ is recommended item $i$*". More concretely, we define a new user embedding as follows:

$$p'_u = \gamma p_u - \epsilon q_i, \text{ where } \epsilon \in \mathbb{R}^+ \text{ and } \gamma \in [0, 1] \tag{4}$$

and hence the new score for the recommendation $(u, i)$ can then be expressed as:

$$s'(u, i) = \langle p'_u, q_i \rangle = \gamma s(u, i) - \epsilon \|q_i\|^2 < s(u, i). \tag{5}$$

For simplicity of exposition, we fix $\gamma = 1$ for now. Intuitively, if we choose a sufficiently large $\epsilon$, we can ensure that the recommended item $i$ is no longer recommended as the score $s'(u, i)$ will drop. Specifically, if we want the new score to be less than $\mathcal{S} \in \mathbb{R}^+$:

$$s'(u, i) = \gamma s(u, i) - \epsilon \|q_i\|^2 < \mathcal{S} \iff \epsilon > \frac{\gamma s(u, i) - \mathcal{S}}{\|q_i\|^2}$$

This leads us to the second step of $\mathcal{C}ontra+$ which is that of using $p'_u$ to determine which items the user would have likely not interacted with. To this end, we construct the explanation set by considering the difference between the old score $s(u, h)$ and the new score $s'(u, h)$ e.g. $\Delta_h = s(u, h) - s'(u, h)$, where $h \in \mathcal{I}_u$. By ordering $\Delta_h$ (for items with a score of at least 4), we assume that the *liked* items that experienced the greatest decrease in score with the new embedding $p'_u$ are the same items that user $u$ would not have interacted with initially. Hence we can state the negative of the antecedent $A$ : "*user $u$ interacted with item $h$*" i.e. $\bar{A}$.

Putting both parts of $\mathcal{C}ontra+$ together we can make the statement $\bar{B}$ : "*We did not recommend item $i$ to user $u$*" and therefore $\bar{A}$ : "*User $u$ would not have interacted with item $h$*". Which is logically equivalent to $A \to B$ i.e. User $u$ interacted with item $h$ and hence we recommended item $i$. We emphasize, that do not claim that we are able to find the one and only explanation, but rather, that we are able to provide a contrapositive explanation which fits our logical statement $\bar{B} \to \bar{A}$, which is equivalent to $A \to B$. This is corroborated by our extensive experiments as well.

To further elucidate our methodology, let us examine a scenario where a user $u$ received a recommendation for the movie *The Godfather II* based on their previous interactions out of which one of them was *The Godfather*. A useful explanation for the user would be the logical statement: "Given your interaction with *The Godfather*, you were recommended *The Godfather II*". Using our proposed contrapositive approach, we generate an explanation by first generating a user embedding $p'_u$ who was not recommended *The Godfather II*. If we then observe that the scores for the previously recommended item such as *The Godfather* significantly decrease compared to the rating provided by the user, we can infer that the absence of the recommendation for *The Godfather II* would likely have been because of the lack of interaction with *The Godfather*. Hence, we can deduce that "If the movie *The Godfather II* was not recommended, the user would not have interacted with *The Godfather*" is logically equivalent to the explanation "*Because you interacted with The Godfather, you were recommended The Godfather II*".

### 3.2 FACTOR MODEL: MLP FACTOR MODELS

Now that we have described the general framework for the SVD model the natural question is how this is applicable to a model for which we do not necessarily have the inner product structure between user and item embeddings $\langle p_u, q_i \rangle$ such as MLP models. In these neural models, even though we are still constructing $p_u, q_i$ i.e. user and item embedding respectively, we no longer compute the inner product but rather concatenate the embeddings before pushing them through multiple MLP layers.

Hence this renders our simple user embedding modification unusable. We therefore propose an alternative method for non-inner product methods which in essence only requires us to backpropagate the scores for a given user to reduce their score for a recommended item $i$. In other words, let $p_u, q_i$ be the user and item embeddings respectively and let $\mathrm{MLP} : \mathbb{R}^{2d} \to \mathbb{R}$ be the MLP that takes as input the concatenation $[p_u, q_i]$ and outputs the corresponding relevance score. In this case, we can update the user embedding $p_u$ over a $k$ iterations as follows, where $\eta$ is a learning rate:

$$p'_u \leftarrow p_u - \eta \nabla_{p_u} \mathrm{MLP}([p_u, q_i]) \tag{6}$$

Note that all the other parameters of the recommender systems remain the same and that we are only modifying the embedding $p_u$. In this case, we again a new user embedding $p'_u$ for user $u$ and are able to repeat the same procedure as above, i.e. select the items that have dropped most in score in the user history based on the new embedding as our explanation set.

We acknowledge that this computation is indeed more computationally heavy as in the SVD case where we were only required to compute the item embedding for item $i$. However, we argue that this computation is significantly smaller than in Tan et al. (2021) as we only require to backpropagate for a single datapoint and user embedding, which is of the same complexity as a forward pass. In our later experiments, we show that our $\mathcal{Contra}+$ for MLP takes less than 1 second, whereas influence function Koh & Liang (2017); Basu et al. (2020) can take at least 5 times longer.

### 3.3 DISCUSSION ON CONTRAPOSITIVE AND COUNTERFACTUAL EXPLANATIONS

Before moving on to our empirical findings, we first need to clearly delineate the differences and similarities between contrapositive and counterfactual explanations within the realm of recommendation systems. These two types of explanations hinge on distinct logical structures. Counterfactual explanations follow a $\bar{A} \to \bar{B}$ logic, while contrapositive explanations adopt a reversed $\bar{B} \to \bar{A}$ logic. In simple terms, counterfactual explanations explore what changes in recommendations $\bar{B}$ occur *upon* the removal of specific elements $\bar{A}$, whereas contrapositive explanations *begin* by noting the changes in recommendations $\bar{B}$, and then seek to identify which elements were removed $\bar{A}$.

To further illustrate these concepts, consider a movie recommendation system. A counterfactual explanation might highlight that removing horror films (the removal $\bar{A}$) from a user's watch history leads to the system no longer recommending thriller movies (the change in recommendation $\bar{B}$). In contrast, a contrapositive explanation begins with an observed change in the recommendation output—say, thriller movies are no longer suggested (the change $\bar{B}$)—and then determines that this change is due to the exclusion of horror films from the user's history (the removal $\bar{A}$). To be clear, these two explanation methods are not exclusive of each other, i.e. a counterfactual explanation could well possibly fulfil the conditions of contrapositive explanations and vice versa. However, the set of explanations is not completely overlapping as is evident from the Toy Example1.1. (A bucket of water can make the road wet instead of rain)

Having established this understanding, we can now turn our attention to the recently proposed concept of *counterfactual backtracking* von Kügelgen et al. (2022), which interestingly shares some parallels with our contrapositive explanations. Traditional counterfactual reasoning, often metaphorically described as creating "*small miracles*", posits hypothetical scenarios where certain features of reality are modified while others persist. Translating this into the recommendation systems domain might entail erasing a segment of a user's history while the remaining part stays unaltered.

However, the backtracking approach diverges from this path. Instead of crafting a new reality, backtracking maintains the laws of the system intact and traces back changes from the outcome to altered initial conditions. In other words, it starts from a change in recommendations and seeks to identify what alterations in the user's history would lead to this new outcome. In this sense, there is an overlap between contrapositive explanations and counterfactual backtracking as both follow a reversed reasoning, tracing back from outcomes to causes.

Both approaches allow us to imagine how varying the user's history would lead to different recommendations. But unlike traditional counterfactual reasoning—which constructs a completely new world by altering the user's history—both contrapositive explanations and counterfactual backtracking keep the laws of the system unaltered and examine how changes in the outcome can be traced back to changes in initial conditions. This distinction offers an intuitively appealing and conceptually novel approach to understanding recommendation systems. Note that von Kügelgen et al. (2022) have not actually proposed a practical algorithm but rather set up a new theoretical framework.

Now that we have thoroughly explored the differences between our proposed methods to conventional counterfactual methods, we move on to the experimental setting. However, it is apparent that different metrics are needed to capture the contrapositive ideas. Hence, we developed a new metric for contrapositive explanations $\mathcal{M}^u$ in recommender systems which we describe in the following.

### 3.4 Contrapositive Explanations Evaluation Metric

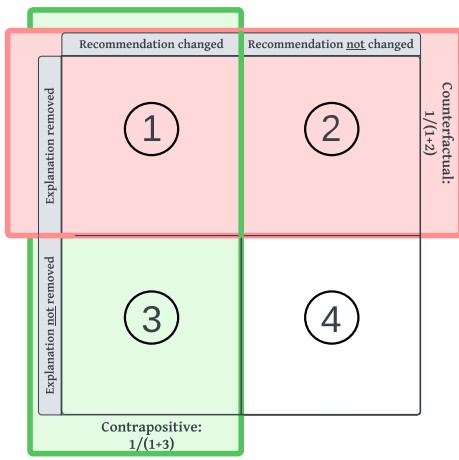

Figure 1: We illustrate how to compute a metric of contrapositive compared to counterfactual. In counterfactual we focus on the top row and compute $1/(1+2)$, whereas in Contrapositive we focus on the left column and compute $1/(1+3)$

In contrast to counterfactual explanations, contrapositive explanations necessitate distinctive evaluation metrics. For counterfactual explanations, performance evaluation typically involves a three-step process: calculating the explanations, removing these explanations from the training data, and verifying whether these alterations changed the recommendation. As depicted in Figure 1, this process corresponds to the top row, where we aim for a high ratio ①/(① + ②) when removing explanations.

Conversely, contrapositive explanations aim to maximize a different ratio: Given a change in recommendation, how many removals instigating this change align with our explanations? This concept is illustrated in the left column of Figure 1, where the desired ratio is ①/(① + ③). Note that these ratios echo the familiar notions of precision and recall prominent in standard machine learning literature, as observed by Watson et al. (2021). However, in this context, we're extending these concepts to fit within the realm of recommender systems.

## 4 Experiments

### 4.1 Experimental Evaluation

To assess the effectiveness of our proposed $\mathcal{C}ontra+$ explanations method, we conducted a series of experiments on well-established benchmark datasets, namely Movielens-100k, Movielens-1M,

and Netflix Harper & Konstan (2015); Bennett et al. (2007). We aim to showcase the versatility of our approach by implementing our contrapositive strategy on two distinct model classes commonly employed in recommender systems: Singular Value Decomposition (SVD) and Multi-Layer Perceptron (MLP) models He et al. (2017). For a comprehensive evaluation, we compared our proposed method against several baseline approaches. Recall, that in this paper we primarily focus on computationally very efficient methods and hence many of the aforementioned methods in the section 2.3 are not comparable due to their computational budget.

**Baselines** The first baseline method, referred to as the Random method, randomly selects explanations from a user's historical data. This method serves as a fundamental sanity check to ensure our contrapositive method outperforms arbitrary selection. The second baseline, the Item Similarity method Yao et al. (2022b), selects historical items most similar to the recommended item as explanations, focusing on similarity-based justifications. This is one of the most commonly used ones as it is computationally very efficient similar to $\mathcal{C}ontra+$. The final baseline for completeness is Influence Function (IF) Koh & Liang (2017), which ascertains explanations based on the historical items with the greatest influence on the recommended item. Note that IF are computationally extremely expensive due to the Hessian matrix. Nevertheless, we believe that IF serves as the gold standard for the other SOTA methods mentioned in section 2.3, which in fact aim to approximate IF.

**Evaluations** Evaluating the quality of explanations generated by our contrapositive method involves using the previously outlined evaluation metric. However, accurately computing this metric requires a more nuanced procedure, which includes the following steps:

Firstly, we sample 10% of each user's historical interactions, denoted as $H_s^u$, and remove them from the training dataset user-item interaction matrix $R$. This process is repeated 100 times per user, yielding 100 models with different subsets $\{H_s^u\}_{s=1}^{100}$ removed from $R$. From these 100 models, we select the subsets $\{H_{\sigma(k)}^u\}_{k=1}^K$ that led to a change in recommendation after retraining (as per the "*recommendation changed*" condition/ left column in Figure 1). Here, $\sigma(k)$ signifies the indexed subset of removals that triggered the recommendation change. We repeat this for 100 users, thus training 10000 models. We emphasise that this retraining is purely for evaluation's sake, the actual explanation method $\mathcal{C}ontra+$ does not require retraining of models. Subsequently, we employ the following metric for contrapositive explanations:

$$\mathcal{M}_{contra} = \frac{1}{n}\sum_{u=1}^n \mathcal{M}^u, \quad \text{where} \quad \mathcal{M}^u = \frac{1}{K}\sum_{k=1}^K \frac{\mathbf{1}(H_{\sigma(k)}^u \cap E_{method})}{|E_{method}|} \tag{7}$$

where $\mathbf{1}$ is the indicator function assessing intersection and $E_{method}$ being the explanation set for a given method. Intuitively, if for every user $u$, the explanations ($E_{method}$) consistently intersect with items causing the recommendations to change ($H_{\sigma(k)}^u$), then the metric $\mathcal{M}_{contra}$ will be high and consequently also the average, thus confirming the usefulness of the contrapositive method.

Lastly, even though, the main goal of this paper is to investigate contrapositive explanations, we also include the counterfactual metric Tran et al. (2021); Yao et al. (2022b) in our experiments for completeness. The counterfactual metric works as follows. For every user, we remove the explanations from the training dataset and subsequently retrain the model. We then compute the ratio of the number of changed recommendations due to the removal of the explanations over the number of users. Intuitively, if this ratio is high, this means that removing the explanations consistently changes the recommendation and hence through the lens of counterfactual logic is considered a good explanation. Given that we are the first to introduce contrapositive explanations to XAI, we believe that, even though tangential, it is important to include the counterfactual metric in order to bridge the gap between the communities.

## 4.2 SVD EXPERIMENTS

To investigate the sensitivity of our method to the model size, we first conducted an ablation study using different latent dimensions for the SVD model on the MovieLens-1M dataset. Top of Figure 2 illustrates the metric $\mathcal{M}_{contra}$ on the $y$-axis for different latent dimensions of $32, 64, 128$. Within each subfigure, we also plot different explanation sizes of $1, 2, 3$ and $5$, along with comparisons to baseline methods. Similarly, we have experiments for the counterfactual metric in bottom Figure 2.

In Figure 2, our $\mathcal{C}ontra+$ firstly demonstrates robustness to changes in latent dimensions and secondly, outperforms the baseline by a significant margin (higher the better) on both evaluation met-

rics, across every latent dimension as well as explanations size. Following this positive result, we extend $\mathcal{C}ontra+$ to two additional datasets: the MovieLens-100k and Netflix datasets.

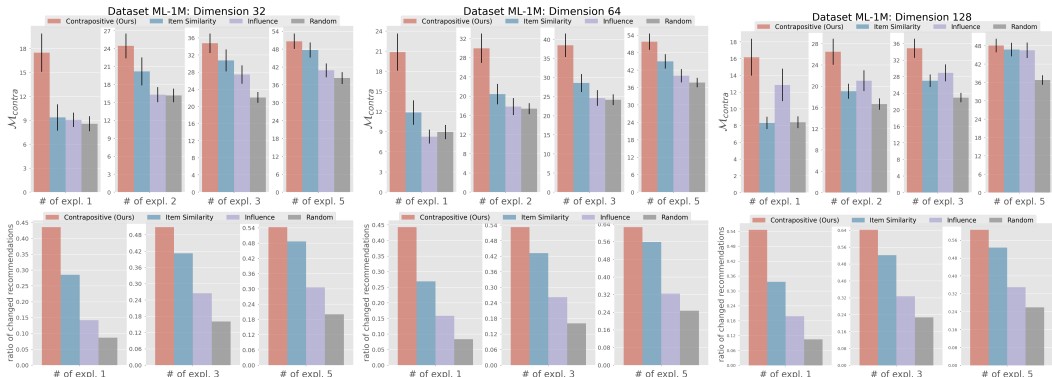

Figure 2: Ablation study with $32$ (left), $64$ (middle), $128$ (right) latent dimensions for the SVD. We compare Contrapositive (Ours) against Item similarity, Influence functions and Random explanation baselines on the $\mathcal{M}_{contra}$ metric (Top) and counterfactual metric (Bottom). (The higher the better)

### 4.2.1 EXPERIMENTS ON ML-1M AND NETFLIX DATASETS

We further examined the efficacy of $\mathcal{C}ontra+$ explanations on the MovieLens-100k and Netflix datasets. Both are widely-used datasets containing approximately $100k$ and $600k$ data points, respectively. Here again, we plot the metric $\mathcal{M}_{contra}$ as well as the counterfactual metric across different explanation sizes. As depicted in Figure 3, our proposed method $\mathcal{C}ontra+$ consistently shows statistically significant improvements over our baseline comparisons across both datasets as well as explanation sizes. It is only for MovieLens-100k, where at explanations size $5$ the item similarity baselines seem to be comparable.

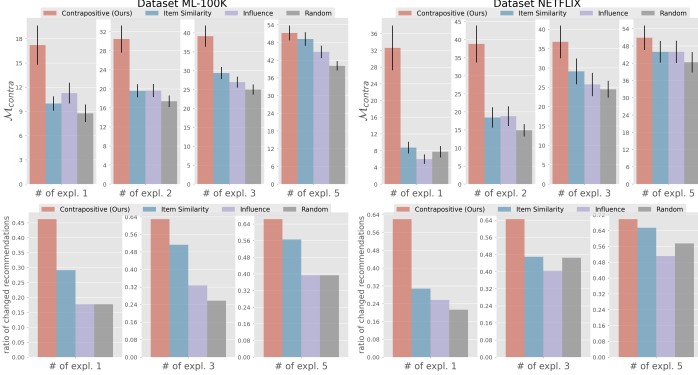

Figure 3: This compares Contrapositive (Ours) against Random, Influence functions and Item similarity baselines on the ML-100K(left) and Netflix (right) datasets. Similarly to the ML-1M dataset, the plots clearly show that our proposed method outperforms baseline methods on both the proposed contrapositive (Top) and counterfactual metric (Bottom). (The higher the better)

### 4.3 MLP EXPERIMENTS

Now that we have established that our methodology works with SVD models, we also perform experiments using MLP models to demonstrate the versatility of our proposed method. In this case, we conducted validation over latent dimensions of $[32, 64, 128]$ and learning rates of $[0.01, 0.001, 0.0001]$, for 3-Layer neural networks and selecting the best model for each dataset based on a held-out validation set. Further details can be found in the Appendix.

Figure 4 compares our proposed method to several baselines across the three different datasets, using the same metric as in the previous experiments. Here again, we see similar or significant improvement in our proposed method compared to the baselines. For ML-100k, we see that $\mathcal{C}ontra+$ is

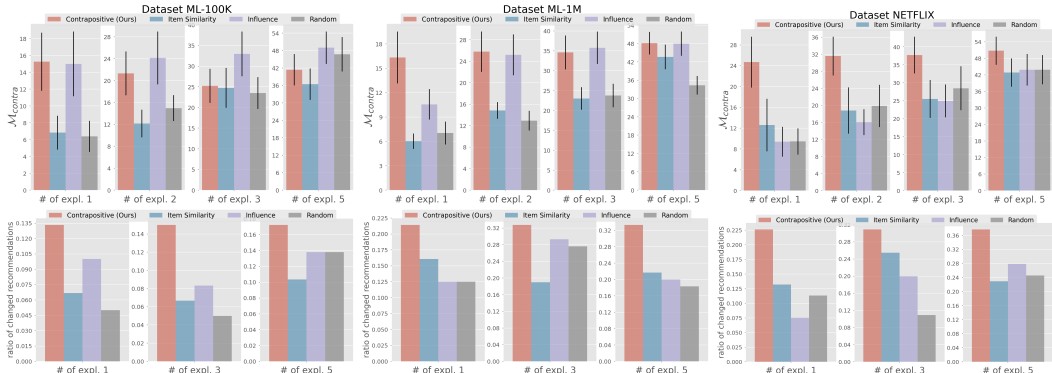

Figure 4: Here we compare Contrapositive (Ours) against Random, Item similarity and Influence baselines on the ML-100K, ML-1M, and Netflix datasets. The plot shows that our method outperforms baseline methods in most of the explanation sizes as well as datasets. (the higher the better)

able to keep up with influence functions for small explanation sizes but seems to be worse on large sizes. In all the remaining experiments, especially the Netflix dataset, our $\mathcal{C}ontra+$ is on par if not better than influence functions on both metrics. Interestingly, $\mathcal{C}ontra+$ performs also very well on the counterfactual metric, which can be explained through Figure 1. Both contrapositive and counterfactual metrics make use of the quantity in the top left corner in Figure 1 in their computation and hence there is a clear correlation between the metrics.

In addition, we would like to emphasize that influence functions were only included in our experiments for the sake of completeness and transparency. As mentioned earlier in Section 2.3, influence functions are computationally expensive due to Hessian computations and are thus not directly comparable to the objective of our paper, which focuses on computationally efficient methods. Nevertheless, we show that even in this case, $\mathcal{C}ontra+$ can perform on par or even better than influence functions, further demonstrating the merits of $\mathcal{C}ontra+$ .

## 5    CONCLUSION, LIMITATIONS AND FUTURE WORK

In this paper, we introduce a novel way to compute explanations, $\mathcal{C}ontra+$ , for recommender systems through the lens of contrapositive logic. The key insight is that the statements $\bar{B} \to \bar{A}$ and $A \to B$ are equivalent. In this case statement, $A$ is "*user interacted with item $j$*" and statement $B$ is "*user was recommended item $i$*". Through extensive examples as well as empirical experiments, we have shown that our proposed method $\mathcal{C}ontra+$ is able to outperform conventional methods on several datasets from a $A \to B$ logic point of view. In addition, we have also shown that our proposed method is computationally much more efficient than methods such as the Influence function, which requires access to the Hessian matrix of a differentiable model. Lastly, we believe that this new way of considering explanations might open up new avenues of research in the field of explainable AI and recommender systems. By using contrapositive logic to compute explanations, we can provide more intuitive explanations, while also improving the efficiency of the computation.

There are however still limitations to our approach, firstly, given that we are unable to *exactly* recover the true data distribution of what caused the model to learn a lower score $s(u, i)$, we are only approximating the negation of the "*did not interact with item $j$*" statement. Even though we have shown how effective our approach is through extensive empirical evidence, more computationally heavy methods on how to properly select the historical items based on the perturbed user embeddings might improve the results. However, we stress that we are primarily interested in computationally efficient methods in this paper and hence leave this interesting avenue for future research. Secondly, we also acknowledge that in the case of Factor models such as SVD the computational complexity of an explanation is of the order of a recommendation, the same cannot necessarily be said for MLP models. In the neural model case, we require a few gradient steps which can lead to higher computational costs. However, this is still significantly cheaper than using methods such as influence functions which are completely unusable for large neural models. Lastly, extensions to different models outside recommender systems such as classifications or regression tasks would be an interesting extension, however, this is outside the scope of this paper and for future work.

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

## A    EXPERIMENTAL DETAILS

Throughout the paper, we had on-and-off access to 50 V100 GPUs with 100 CPUs.

### A.1    SVD MODELS

For our SVD model, we use the `Surprise` Hug (2020) library in Python which allows us to quickly train SVD models on datasets such as the MoiveLens 100K, MovieLens 1M and Netlfix dataset. As mentioned in the main text, we compute the metric by firstly sampling 10% of each user's historical interactions, denoted as $H_s^u$, and remove them from the training dataset user-item interaction matrix $R$. This process is repeated 100 times per user, yielding 100 models with different subsets $\{H_s^u\}_{s=1}^{100}$ removed from $R$. From these 100 models, we select the subsets $\{H_{\sigma(k)}^u\}_{k=1}^{K}$ that led to a change in recommendation after retraining (as per the "*recommendation changed*" condition/ left column in Figure 1). Here, $\sigma(k)$ signifies the indexed subset of removals that triggered the recommendation change. We repeat this for 100 users, thus training 10000 models. We emphasise that this retraining is purely for evaluation's sake, the actual explanation method $\mathcal{Contra}+$ does not require retraining of models. Subsequently, we employ the following metric for contrapositive explanations:

$$\mathcal{M}_{contra} = \frac{1}{n}\sum_{u=1}^{n}\mathcal{M}^u, \quad \text{where} \quad \mathcal{M}^u = \frac{1}{K}\sum_{k=1}^{K}\frac{\mathbf{1}(H_{\sigma(k)}^u \cap E_{method})}{|E_{method}|} \tag{8}$$

where $\mathbf{1}$ is the indicator function assessing intersection and $E_{method}$ being the explanation set for a given method. Intuitively speaking, if for every user $u$, the explanations ($E_{method}$) consistently intersect with items causing the recommendations to change ($H_{\sigma(k)}^u$), then the metric $\mathcal{M}_{contra}$ will be high and consequently also the average, thus confirming the usefulness of the contrapositive method. By measuring, how many times our explanations are intersecting with the items that cause the recommendations to change, we are in fact in the first column of figure 1 i.e. conditioned on the recommendation having changed, was the explanations removed? For this reason, this is a good metric compared to existing counterfactual metrics. Note that the sum of the four squares always has to add up to the same number and hence row 1 completely determines row 2 and similarly for the columns.

We would also like to note that, this metric indeed is hard to compute, however, only has to be done for validation and not inference. Future work will consider possible surrogates to this metric i.e. for example using the counterfactual metric for instance.

For the SVD model, we also have hyperparameters to tune which is that of $\epsilon$ and $\gamma$ i.e.

$$p_u' = \gamma p_u - \epsilon q_i, \text{ where } \epsilon \in \mathbb{R}^+ \text{ and } \gamma \in [0,1] \tag{9}$$

In order to determine which hyperparameter fits best, compute the metric $\mathcal{M}_{contra}$ on 20 held-out users and pick the corresponding $\gamma = [0, 0.1, 0.2, ..., 1.0]$ and $\epsilon$ accordingly. Note that $\epsilon$ is defined through $\mathcal{S} = [1, 2, 3, 4]$ and hence searching through $\epsilon$ is searching through $\mathcal{S}$, i.e. recall:

$$s'(u,i) = \gamma s(u,i) - \epsilon\|q_i\|^2 < \mathcal{S} \iff \epsilon > \frac{\gamma s(u,i) - \mathcal{S}}{\|q_i\|^2}$$

Note if $\gamma = 1$ we in fact recover the item similarity baseline.

### A.2    MLP MODELS

For MLP models we employ the same metrics as in SVD models. However, for the MLP model, we have a few more hyperparameters that we need to tune. Recall in the formulation of the MLP models that we update the user embedding $p_u$ over a $k$ iterations, where $\eta$ is a learning rate:

$$p_u' \leftarrow p_u - \eta\nabla_{p_u}\text{MLP}([p_u, q_i]) \tag{10}$$

In order to determine which hyperparameter fits compute the metric $\mathcal{M}_{contra}$ on 20 held-out users. Once, we have them we can then pick the hyperparameters $k$ as well as $\eta$ appropriately. The ranges that we checked for $k = [500, 1000, 2000]$ and $\eta = [0.0001, 0.0005]$. Experimentally, we note that our method does not seem to be sensitive to $k$ as long as the resulting score for using $p_u'$ is below 4 i.e. not recommended anymore. Interestingly higher $\eta$ do not seem to work well. We hypothesise that the embedding changes too quickly and hence is no longer close to the original users.

