# OpenReview forum: "Explaining recommendation systems through contrapositive perturbations"
_ICLR.cc/2024/Conference — Submitted to ICLR 2024_

### Official Review · Reviewer_ck6L · 2023-10-31

**Soundness:** 2 fair
**Presentation:** 3 good
**Contribution:** 2 fair
**Rating:** 3
**Confidence:** 4

**Summary:**

This paper considers the interpretability problem “because the user interacted with item $j$, we recommend item $i$ to the user” in a factorization model commonly used in recommender systems. From the perspective of contrapositive logit (“because the user did not interact with item $j$, we did not recommend item $i$ to the user”), this paper proposes a new explanation algorithm (Contra+) consisting of two steps: (1) perturbing the user embedding to ensure item $i$ is not recommended; (2) given the perturbed user embedding, identifying the historical items that have lost most relevance to the user. Overall, the proposed algorithm is interesting but is more empirical and lacks theoretical guarantees.

**Strengths:**

1. the writing is well and the presentation is clear.

2. the topic is interesting.

**Weaknesses:**

1. the proposed method is more empirical and lacks theoretical guarantees (see main question 1 for more details).

2. something about the key logic of the algorithm is not clearly explained (see main question 2 for more details).

**Questions:**

**Main Questions**

1.	the perspective of contrapositive logit is not fully novel. In fact, Pearl (1999)[1] defined the notation of **probability of necessary causation**, which follows the same logic as contrapositive. There may be some connection between the probability of necessary causation and the method proposed in this paper. Linking the method proposed in this paper with the necessary causality probability may provide a theoretical guarantee for the method proposed in this paper. Could you discuss something about the possible connections?

2.	Here are some questions about the key logic of the proposed method.

>(1) In terms of the perturbation, (a) Why only the user embedding is perturbed and not the item embedding? It is a bit confusing to me. intuitively, the user after the perturbation is no longer the same user before. (b) Do all user-item pairs, using the same strength ($\gamma$ and $\epsilon$ in equation (4)) of perturbation? (c) How to choose the parameters $\gamma$ and $\epsilon$ in practical applications?

>(2) For step 2, i.e., identifying the historical items that have lost most relevance to the user. Why the historical items that have lost the most relevance to the new perturbed user embedding is the explanation? Is it equivalent to the statement “because the user did not interact with item $j$, we did not recommend item $i$ to the user”?

[1] Judea Pearl (1999), Probabilities of causation: three counterfactual interpretations and their identification.


**Minor Questions**

(1)	There are some problems with the format of the citation. For example, at the end of the first paragraph in the Introduction, the citation format should appear as (Lu et al., 2012; Aggarwal et al., 2016; Beel et al., 2016; Jannach et al., 2022), which can be generated using the \citep{XXX} command.

(2)	There are some grammatical errors. For example, at the end of the Abstract, “… because the user did not **interacted** with item $j$ ….” should be  “… because the user did not **interact** with item $j$ ….”.

---

> ### Author Response · Authors · 2023-11-17
> **Part [1/2] Rebuttal "Explaining recommendation systems through contrapositive perturbations"**
>
> Thank you! Thank you very much for your insightful comment regarding the connection between the contrapositive logic in our Contra+ method and Judea Pearl’s concepts of the probability of necessary and sufficient causation. This is indeed a very interesting connection to causality that we have tried to touch upon in our discussion on connecting our work to Counterfactual backtracking [1]. We appreciate this opportunity to clarify and elaborate on the novel aspects of our approach, particularly on the established frameworks of causal inference.
>
> 1. Connection Between Contra+ and Pearl’s Framework of Necessity and Sufficiency
>
> The Contra+ method, while distinct in its application, shares a conceptual connection with Judea Pearl’s framework of necessity and sufficiency in causal inference. This connection lies in the underlying logic that both frameworks employ.
>
> - Logical Foundations (our approach): In propositional logic, necessity and sufficiency are expressed as x → y (sufficiency) and y → x (necessity). The Contra+ method utilises contrapositive logic, where the inversion of these implications (-y → -x for sufficiency and -x → -y for necessity) is central to its explanatory mechanism. For instance, if a recommendation system suggests a movie based on a user's preference for a genre, our method uses contrapositive logic to infer non-preferences from non-recommendations, akin to the logical structure of sufficiency.
>
> - Probabilistic causal Interpretation: Pearl’s work extends these concepts into the probabilistic and causal realm, assessing the likelihood of causation. While Contra+ does not directly engage with probabilistic causation, its logical structure echoes the principles Pearl discusses. The method’s efficiency in explanation generation can be seen as a practical application of these ideas, albeit in a different context and using different assumptions as we will discuss in the following.
>
> 2. Delineating the Difference Between Our Method and Pearl’s Framework
> While there are conceptual similarities, there are key differences in the application and objectives of our Contra+ method and Pearl’s causal inference framework.
> - Focus on Explanation vs. Causal Analysis: Pearl’s framework is deeply rooted in understanding and quantifying causal relationships in complex systems, where one has to assume a causal structure of the problem. In contrast, our Contra+ method is primarily focused on enhancing the interpretability and efficiency of recommendations in factorization models without any causal assumptions. Our approach is about using logical inferences to provide clear and computationally feasible explanations for recommendations, not to establish causal relationships as this would require causal assumptions such as the graph (not clear in our case) etc. Nevertheless, we agree that this future research direction of bridging this gap between these two frameworks is interesting and would allow us to potentially provide theoretical guarantees.
>
> 3. Future Work Outside the Scope of This Paper
> Nevertheless, there are several avenues for future research that, while outside the scope of this paper, could enrich the Contra+ method.
> - Integrating Causal Inference Principles: One promising direction is the exploration of how causal inference principles can be applied to explaining recommendation systems. This could involve adapting aspects of Pearl’s framework to better understand the causal mechanisms behind user preferences and behaviours, thereby enhancing the depth and robustness of our explanations. We have also, in section 3.3, mentioned a connection to the recent paper on "*Counterfactual backtracking*" [1] which we believe gives a theoretical understanding of what Contra+ is doing under the hood. However, given that we wish to have a general algorithm without too many causal assumptions, we did not opt to connect them via this route 1-to-1. However, for future work, which is outside the scope of this paper, we will consider making this connection more formal as it would enhance the reliability of the method. This paper primarily focuses on a computationally efficient manner to generate explanations for recommender systems and we have shows on extensive experiments the merits of our proposed method.
>
> In conclusion, while the Contra+ method shares logical underpinnings with Pearl’s framework of necessity and sufficiency, it stands as a distinct approach focused on the specific challenges of explanation in recommendation systems. Future work would aim to explore these theoretical connections further, expanding the method’s capabilities and theoretical foundations.
>
> [1] von Kügelgen, J, Counterfactual Backtracking

---

> > ### Author Response · Authors · 2023-11-17
> > **Part [2/2] Rebuttal "Explaining recommendation systems through contrapositive perturbations"**
> >
> > > 2. Here are some questions about the key logic of the proposed method.
> >
> > > (1) In terms of the perturbation, (a) Why only the user embedding is perturbed and not the item embedding? It is a bit confusing to me. intuitively, the user after the perturbation is no longer the same user before. (b) Do all user-item pairs, using the same strength ($\gamma$ and $\epsilon$ in equation (4)) of perturbation? (c) How to choose the parameters $\gamma$ and $\epsilon$ in practical applications?
> >
> > > (2) For step 2, i.e., identifying the historical items that have lost most relevance to the user. Why the historical items that have lost the most relevance to the new perturbed user embedding is the explanation? Is it equivalent to the statement “because the user did not interact with item j, we did not recommend item i to the user”?
> >
> > Firstly, we thank the reviewer for these interesting and thought-provoking questions.
> > For the first question, we opted to perturb the embedding of the users instead of the embeddings of the items, because we wanted to consider the case where only the recommendation of a single person changes. If we were to change the item embedding we would fall into two potential problems.
> > 1. We would not only change the recommendation of the user that we are interested in but also all other users that have or will interact with this item, as the item embeddings are shared.
> > 2. By changing the item embedding, we do not have a canonical way to translate this change to the user's historical items, whereas, with the changes in the user embedding, we have a natural way (computing the scores with the new user embedding).
> > Next in terms of whether our hyperparameters \epsilon and \gamma, were chosen using held-out users as we described in the appendix. The strength of these parameters was fixed throughout user-item pairs for simplicity. As can be seen through our experiments, this simple but effective perturbation on the user embeddings gives significant performance gains on both the contrapositive as well as the counterfactual baselines. Future work would consider how these might be adaptable for a given user-item pair.
> >
> > For the second question, we argue that because the item has lost significantly in terms of scores according to the new embedding, the user would not have interacted with this item in the first place. Note that we have only considered items of score 4 or above to be explanations as these would be generally considered to be items that the user liked [sec 3.1]. However, if the new embedding suggests that the user would rate this item significantly lower than 4, in our paper, we hypothesize that this corresponds to "the user did not interact with the item ". We empirically back this claim up in our experimental section, where we show that our proposed logic/hypothesis can indeed capture contrapositive logic efficiently.
> >
> > We thank the reviewer again for the thought-provoking questions to improve our paper. We believe that we have answered all your questions and hope that the reviewer will consider raising their score. If there are any other questions, we would be more than happy to clarify.
> >
> > [1] von Kügelgen, J, Counterfactual Backtracking

---

### Official Review · Reviewer_PpEb · 2023-11-01

**Soundness:** 3 good
**Presentation:** 2 fair
**Contribution:** 2 fair
**Rating:** 5
**Confidence:** 3

**Summary:**

This paper is trying to address the challenge of explaining recommendations, which is meaningful and important because recommender systems like factorization models based or neural network based are lack of transparency. The paper introduces a novel approach called "contrapositive explanations (Contra+)" to provide clear and efficient explanations for recommendations. Contra+ focuses on finding explanations in the form of "Because the user interacted with item j, we recommend item i to the user." This is in contrast to traditional counterfactual explanations, which aim to explain why an item was not recommended. This paper provides detailed discussion for previous methods, many toy examples and figures to make the concepts easier for readers to understand. Finally, the authors demonstrate the effectiveness and efficiency of Contra+ through empirical experiments on real-world datasets.

**Strengths:**

S1: This paper considers a interesting questions (explaining recommendation system) from a contrapositive perspective, which is novel.

S2: This paper provides detailed discussion for previous methods, many toy examples and figures to make the concepts easier for readers to understand.

S3: This paper gives a comprehensive review of differences and similarities between contrapositive and counterfactual explanations.

**Weaknesses:**

W1: The key concern is whether there is another way to get the "explanation". Further, is there infinite number of ways to perturbation the embedding that can achieve the same purpose, i.e., "We did not recommend item i to user u"? In such case, does each way of perturbing embedding correspond to a different h, i.e., "User u would not have interacted with item h"? How can we distinguish merits and drawbacks of each perturbation?

W2: Previous literature like Tan et al. [1], studied cause on a particular aspect, i.e., If the item had been slightly worse on [aspect(s)], then it will not be recommended. This can find the cause on a particular aspect, whereas in this paper, the cause is found on perturbation on all embedding. Is any comments for the difference?

W3: The authors give a lot of toy examples, such as rain and slippery roads, or godfather and godfather 2. Can some experiments be added to give some examples of real world datasets where the proposed method finds an explanation? For example, in Netflix or ML-1M, are there any cases where users don't interact with "computer" because "cell phone" is not suggested?

W4: Counterfactual explanations don't necessarily guarantee removing the explanation or changing the recommendation. Therefore, in figure 1, counterfactual explanations should be 1 as a proportion of all areas, that is, 1/(1+2+3+4), not 1/(1+2).

W5: The experiment process is Evaluations part is not so clear. For example, why is $M_{contra}$ greater than 1? In addition, consider doing some runtime experiments and some other hyper-parameter sensitivity analysis or in-depth analysis like the effect of varying total amount data could be better.

[1] Juntao Tan, Shuyuan Xu, Yingqiang Ge, Yunqi Li, Xu Chen, and Yongfeng Zhang. Counterfactual explainable recommendation. In Proceedings of the 30th ACM International Conference on Information & Knowledge Management, pp. 1784–1793, 2021

**Questions:**

Please refer to the weaknesses part for the questions.

=== AFTER REBUTTAL ===

I thank the authors for taking the time to answer my questions, which addresses some of my concerns. However, I still have some concerns about the motivation and methodology. Hence, I may maintain my score.

---

> ### Author Response · Authors · 2023-11-17
> **Part [1/2] Rebuttal "Explaining recommendation systems through contrapositive perturbations"**
>
> First of all, we would like to thank the reviewer for their comprehensive review to improve our paper as well as allowing us an opportunity to clarify any concerns or misunderstandings. We would also like to thank the reviewer for praising our paper for considering an "*interesting question (explaining recommendation system) from a contrapositive perspective, which is novel.*" Here below, we have addressed all your concerns.
>
> > W1: The key concern is whether there is another way to get the "explanation". Further, is there infinite number of ways to perturbation the embedding that can achieve the same purpose, i.e., "We did not recommend item i to user u"? In such case, does each way of perturbing embedding correspond to a different h, i.e., "User u would not have interacted with item h"? How can we distinguish merits and drawbacks of each perturbation?
>
> We thank the reviewer for this clarification question. We agree and have acknowledged this observation in our main text of the paper : "We emphasize, that do not claim that we are able to find the one and only explanation, but rather, that we are able to provide a contrapositive explanation which fits our logical statement -B→ -A, which is equivalent to A → B. This is corroborated by our extensive experiments as well."
> To provide more details, even though there are in theory an infinite number of ways to perturb the representations, in this paper, our main focus is on computationally efficient ways to construct contrapositive explanations. We could use more elaborate ways to construct perturbations, however, given that we are practically oriented, we decided to go with the most natural and efficient way to construct the perturbations, which is our proposed method Contra+. We show the merits of Contra+ on numerous experiments in section 4 and demonstrate that this way of constructing perturbations is effective and even matches the performance of the Influence function in some cases. Note that influence functions, which are computationally much more expensive have only been added for completeness and transparency. Future research would investigate how one could trade off computational efficiency with possibly superior explanations. However, this is outside the scope of this paper as we are mainly focusing on computationally efficient algorithms.
>
> > W2: Previous literature like Tan et al. [1], studied cause on a particular aspect, i.e., If the item had been slightly worse on [aspect(s)], then it will not be recommended. This can find the cause on a particular aspect, whereas in this paper, the cause is found on perturbation on all embedding. Is any comments for the difference?
>
> In Tan et al. the authors focus on counterfactual explanations (-A -> -B i.e. if A did happen then B would not happen), which are different to our proposed method, which utilizes the contrapositive (-B -> -A) nature of explanations. However, the reviewer is raising a very interesting point here, whereby one could potentially perturb the embeddings only in a specific direction such that only a given aspect can be affected and subsequently the rating will drop (-B). This perturbation could then be used to determine the contrapositive explanations. However, there are also challenges associated with this approach, such as accurately determining the appropriate direction to push the embeddings in the direction of these aspects.
> Additionally, it remains uncertain whether this approach would be more effective compared to the perturbation suggested in our paper, which showcased substantial enhancements in our experimental section. Note that in our contrapositive formulation, we only care about -B→ -A, i.e. conditioned on the user not being recommended item j, the user would not have watched item i. Therefore, any method of inducing -B could be used and we leave this interesting avenue for future research as this would be out of the scope of this paper.
> In summary, we thank the reviewer for their interesting suggestion to incorporate aspect-specific perturbation in the embedding space, a concept that could provide a valuable addition to our paper. However, it is important to acknowledge that our unique contribution lies in the explanation from a different perspective, which is separate from aspect-specific perturbation.
>
> [1] Kaffes et al Model-Agnostic Counterfactual Explanations of Recommendations
>
> [2] Yao et al, Counterfactually Evaluating Explanations in Recommender Systems
>
> [3] Tran et al, Counterfactual Explanations for Neural Recommenders

---

> > ### Author Response · Authors · 2023-11-17
> > **Part [2/2] Rebuttal "Explaining recommendation systems through contrapositive perturbations"**
> >
> > > W3: The authors give a lot of toy examples, such as rain and slippery roads, or godfather and godfather 2. Can some experiments be added to give some examples of real world datasets where the proposed method finds an explanation? For example, in Netflix or ML-1M, are there any cases where users don't interact with "computer" because "cell phone" is not suggested?
> >
> > We appreciate the reviewer's comments on this, instead of illustrating the explanations on a few random examples, we instead opted to illustrate a more holistic and objective metric on how well the proposed method works in our experimental results. We believe, just like in previous work [1, 2], that average metrics would be more informative to back up our claims. Nevertheless, upon the reviewer's request, we did look into some examples of explanations and have found the following interesting one for which "The Godfather" was recommended. In this case, given that our recommender systems do not take into account the tags, it is interesting that Contra+ was the only method to pick a drama as an explanation in this case:
> >
> > Recommendation: 858, "Godfather, The (1972)",Crime|Drama
> > Contra+ explanations: Before Night Falls (2000), Drama
> > Influence explanations: Twelve Monkeys (a.k.a. 12 Monkeys) (1995), Mystery|Sci-Fi|Thriller
> > Random explanations: Raiders of the Lost Ark (Indiana Jones and the Raiders of the Lost Ark) (1981),Action|Adventure
> >
> > > W4: Counterfactual explanations don't necessarily guarantee removing the explanation or changing the recommendation. Therefore, in figure 1, counterfactual explanations should be 1 as a proportion of all areas, that is, 1/(1+2+3+4), not 1/(1+2).
> >
> > It is indeed true that in general "*counterfactual explanations*" do not necessarily guarantee to change the recommendation. The goal of figure 1 was to illustrate that, in standard counterfactual metrics as used in previous papers [1, 2, 3], we start by always removing the counterfactual explanations, i.e. conditioning on the first row in Figure 1. Subsequently, we want to investigate how many times the recommendations changed after the removal of the explanations. Hence, given the way the counterfactual metric is computed in the literature, the second row, i.e. quadrants 3 and 4 are zero.
> >
> > "*The counterfactual metric works as follows. For every user, we remove the explanations from the training dataset and subsequently retrain the model. We then compute the ratio of the number of changed recommendations due to the removal of the explanations over the number of users. Intuitively, if this ratio is high, this means that removing the explanations consistently changes the recommendation and hence through the lens of counterfactual logic is considered a good explanation.*"
> > We thank the reviewer for pointing out this confusion and we will amend this in the final version of our paper.
> >
> > > W5: The experiment process is Evaluations part is not so clear. For example, why is $\mathcal{M}_{contra}$ greater than 1? In addition, consider doing some runtime experiments and some other hyper-parameter sensitivity analysis or in-depth analysis like the effect of varying total amount data could be better.
> >
> > We thank the reviewer for their careful reading of our paper. The numbers are percentages which we unfortunately forgot to update in the figure. We thank the reviewer again for their constructive comment to improve our paper. In terms of runtime experiments, we have additionally collected the runtimes and noticed that for all methods except the influence functions, the run times are < 1e-2 seconds whereas influence functions take on average up to 4 seconds
> >
> >
> > Lastly, we hope that the above has addressed all of the reviewer's questions and clarified any misunderstandings in our work. Hence, we hope that the reviewer will consider raising their score if the above has appropriately answered your questions. If you have any other questions, we are more than happy to clarify any further questions.
> >
> > [1] Kaffes et al Model-Agnostic Counterfactual Explanations of Recommendations
> > [2] Yao et al, Counterfactually Evaluating Explanations in Recommender Systems
> > [3] Tran et al, Counterfactual Explanations for Neural Recommenders

---

### Official Review · Reviewer_5dmR · 2023-11-06

**Soundness:** 2 fair
**Presentation:** 3 good
**Contribution:** 3 good
**Rating:** 5
**Confidence:** 2

**Summary:**

The paper proposes an interesting explanation method to explain recommendation systems through contrapositive perturbations, leveraging the key insight that (negation of B => negation of A) and (A=>B) are equivalent . The proposed method is computational efficient to SVD and MLP-based recommender systems. Lastly, the paper evaluates the approach against benchmarks on several datasets to demonstrate its effectiveness and efficiency in explanations.

The approach seems novel and interesting but have some questions and concerns on the experimentation session.  Mostly concern if the paper is comparing to the compelling baselines, and M_contra seems to on part to "influence" functions in some datasets:
Q1: do we have compelling baselines to compare against? The reason asked is because if we comparing item similarity and influence comparing to random, they seem to be not very statistically different in M_contra in many cases (i.,e, Figure 2 on Dimension 32 for # of expl Q2:  in Figure 4, it seems that "Influence" is comparable or have higher M_contra value as "Contrapositive" approach in Dataset ML-100k, is that expected?

**Strengths:**

The paper proposes an interesting explanation method to explain recommendation systems through contrapositive perturbations, leveraging the key insight that (negation of B => negation of A) and (A=>B) are equivalent . The proposed method is computational efficient to SVD and MLP-based recommender systems. Lastly, the paper evaluates the approach against benchmarks on several datasets to demonstrate its effectiveness and efficiency in explanations.

**Weaknesses:**

Mostly have some concern and/or questions on the Experiment session if the paper is comparing to the compelling baselines.
Q1: do we have compelling baselines to compare against? The reason asked is because if we comparing item similarity and influence comparing to random, they seem to be not very statistically different in M_contra in many cases (i.,e, Figure 2 on Dimension 32 for # of expl Q2:  in Figure 4, it seems that "Influence" is comparable or have higher M_contra value as "Contrapositive" approach in Dataset ML-100k, is that expected?

**Questions:**

Mostly have some concern and/or questions on the Experiment session to prove out on the claims.
Q1: in Figure (2) and (3), as the number pf experiments increase, in particular at 5, it seems that the contrapositive approach is non-stats sign from other baselines, especially Item Similarity or Influence. Was this the expected behavior?
Q2: in Figure 4,  it seems that "Influence" is comparable or have higher M_contra value as "Contrapositive" approach in Dataset ML-100k, is that expected.

---

> ### Author Response · Authors · 2023-11-17
> **Rebuttal for "Explaining recommendation systems through contrapositive perturbations"**
>
> First of all, we would like to thank the reviewer for their time reviewing our manuscript and giving thought-provoking clarification questions. We would also like to thank the reviewer for praising our paper for being an "*interesting explanation method to explain recommendation systems through contrapositive perturbations*". In the following, we will address and clarify all the questions raised in the review.
>
> > Q1: in Figure (2) and (3), as the number pf experiments increase, in particular at 5, it seems that the contrapositive approach is non-stats sign from other baselines, especially Item Similarity or Influence. Was this the expected behavior?
>
> We thank the reviewer for pointing out this interesting experimental observation in our paper. The fact that the results are similar to influence function and item similarity for ML-100K, when the number of explanations increases to 5 is somewhat expected because the contrapositive and the counterfactual metrics are correlated. This has been shown in the experimental section as well as mentioned at the bottom of section 4, where we state that "*Both contrapositive and counterfactual metrics make use of the quantity in the top left corner in Figure 1 in their computation and hence there is a clear correlation between the metrics*". Hence, as we increase the number of explanations, quadrant 1 will increase and quadrants 2 and 3 will decrease for the counterfactual and contrapositive metric respectively.
>
> Intuitively, as we increase the number of explanations, we are more likely from the counterfactual perspective, to select a historical item that, if removed, will change the recommendation. Therefore, as we increase the number of explanations, we expect both counterfactual and contrapositive metrics to increase up to a certain point and we observe that at around 5 explanations they both plateau for ML-100K.
>
> Lastly, we want to emphasize that in real application settings (such as social media apps), the goal is to keep the explanations to a minimum to reduce user confusion. Hence, we only added the experiments with 5 explanations as ablation to understand the behaviour and large explanation sizes. Previous works such as [1] only considered explanation sizes of 3.
>
>
> > Q2: in Figure 4, it seems that "Influence" is comparable or have higher M_contra value as "Contrapositive" approach in Dataset ML-100k, is that expected.
>
> We thank the reviewer for this observation and we will add a detailed discussion in the final version of the paper. In particular, we want to first highlight that influence functions have only been added for completeness and transparency, as they are not computationally feasible in large recommender systems. The main focus of our paper is on computationally efficient methods for explanations in the realm of recommender systems. Therefore computational cost of influence functions is not directly comparable, as influence functions require the computation of a Hessian matrix inverse. Nevertheless, for completeness and transparency, we show that even despite these computational differences, our proposed method Contra+ can perform on par with influence functions, not only on the contrapositive but also on counterfactual metric, which is a very promising result.
> Secondly, in terms of the results being higher or comparable to our proposed method, we would like to highlight that for all the methods that are computationally comparable to our proposed methods (Random, item similarity), we perform significantly better for explanation sizes 1 to 3 for the contrapositive metric and do significantly better on the counterfactual metric across explanation sizes.
>
> We thank the reviewer again for their time reviewing our paper and hope that the above has clarified all of the reviewer's questions. We would appreciate it if the reviewer could increase their score if the above has appropriately answered all your concerns and we are more than happy to answer any remaining questions the reviewer might have.
>
> [1] Yao et al, Counterfactually Evaluating Explanations in Recommender Systems

---

### Meta-Review · Area_Chair_hQ5R · 2023-12-06

**Metareview:**

In this paper, the introduce a novel way to compute explanations for recommender systems which they call "Contra+ "
Their key insight is that the statements B¯ → A¯ and A → B are equivalent where in this analysis statement A is “user interacted with item j” and statement B is “user was recommended item i”.   The proposed method is computational efficient to SVD and MLP-based recommender systems. The paper evaluates the approach against benchmarks on several datasets to demonstrate its effectiveness and efficiency in explanations.

Strengths:
-This paper considers a interesting questions (explaining recommendation system) from a contrapositive perspective, which is novel.

-This paper provides detailed discussion for previous methods, many toy examples and figures to make the concepts easier for readers to understand.

- This paper gives a comprehensive review of differences and similarities between contrapositive and counterfactual explanations.

Weaknesses:

There are some questions and concerns on the experimentation session. Mostly  if the paper is comparing to the compelling baselines,
because  similarity and influence when compared to random, do not seem to be not very statistically different in many cases (i.,e, Figure 2 on Dimension 32 for # of expl Q2: in Figure 4, it seems that "Influence" is comparable or have higher M_contra value as "Contrapositive" approach in Dataset ML-100k.

It would have been interesting to compare the reasons given by the contrapositive system to something like counterfactual backtracking as this seems most similar.  How often do contrapositive and counterfactual methods return the same explanation?  The authors emphasize that their method will return a possible explanation, but it will be one of many.  They do not test the same way that counterfactual reasoning does.

"We emphasize, that do not claim that we are able to find the one and only explanation, but rather, that we are able to provide a contrapositive explanation which fits our logical statement -B→ -A, which is equivalent to A → B. This is corroborated by our extensive experiments as well."

I wished the authors had explained their experiments better.  I looked into "Counterfactual Explanations for Neural Recommenders"
Khanh Hiep Tran, Azin Ghazimatin, Rishiraj Saha Roy to try to understand the evaluation.  I wish the authors had reported their results in the same way as Tran et al and that they had explicitly written out their algorithm the way Tran et al did.  I also wish the authors had made it clear that the result labeled as "Influence" was the ACCENT counterfactual model by Tran et al.

**Justification For Why Not Higher Score:**

I agree with the reviewers that the analysis does not seem to compare against a challenging baseline.

**Justification For Why Not Lower Score:**

The paper seems to have some merit, especially with respect to the advantages of the computational efficiency and explainability of recommendations.

---

### Decision · Program_Chairs · 2024-01-16

Reject